# TOWARDS EVALUATING THE ROBUSTNESS OF NEURAL NETWORKS LEARNED BY TRANSDUCTION

**Jiefeng Chen** [1]  **Xi Wu** [2]  **Yang Guo** [1]  **Yingyu Liang** [1]  **Somesh Jha** [1,3]
[1] University of Wisconsin-Madison   [2] Google   [3] XaiPient
{jiefeng, yguo, yliang, jha}@cs.wisc.edu   wuxi@google.com

## ABSTRACT

There has been an emerging interest in using transductive learning for adversarial robustness (Goldwasser et al., NeurIPS 2020; Wu et al., ICML 2020; Wang et al., ArXiv 2021). Compared to traditional defenses, these defense mechanisms "dynamically learn" the model based on test-time input; and theoretically, attacking these defenses reduces to solving a bilevel optimization problem, which poses difficulty in crafting adaptive attacks. In this paper, we examine these defense mechanisms from a principled threat analysis perspective. We formulate and analyze threat models for transductive-learning based defenses, and point out important subtleties. We propose *the principle of attacking model space* for solving bilevel attack objectives, and present Greedy Model Space Attack (GMSA), an attack framework that can serve as a new baseline for evaluating transductive-learning based defenses. Through systematic evaluation, we show that GMSA, even with weak instantiations, can break previous transductive-learning based defenses, which were resilient to previous attacks, such as AutoAttack. On the positive side, we report a somewhat surprising empirical result of "*transductive adversarial training*": Adversarially retraining the model using fresh randomness at the test time gives a significant increase in robustness against attacks we consider.

## 1 INTRODUCTION

Adversarial robustness of deep learning models has received significant attention in recent years (see Kolter & Madry (2018) and references therein). The classic threat model of adversarial robustness considers an *inductive setting* where a model is learned at the training time and fixed, and then at the test time, an attacker attempts to thwart the fixed model with adversarially perturbed input. This gives rise to the adversarial training (Madry et al., 2018; Sinha et al., 2018; Schmidt et al., 2018; Carmon et al., 2019) to enhance adversarial robustness.

Going beyond the inductive threat model, there has been an emerging interest in using *transductive learning* (Vapnik, 1998)[1] for adversarial robustness (Goldwasser et al., 2020; Wu et al., 2020b; Wang et al., 2021). In essence, these defenses attempt to leverage *a batch* of *test-time inputs*, which is common for ML pipelines deployed with batch predictions (bat, 2021), to *learn an updated model*. The hope is that this "test-time learning" may be useful for adversarial robustness since the defender can adapt the model to the perturbed input from the adversary, which is distinct from the inductive threat model where a model is fixed after training.

This paper examines these defenses from a principled threat analysis perspective. We first formulate and analyze rigorous threat models. Our basic 1-round threat model considers a single-round game between the attacker and the defender. Roughly speaking, the attacker uses an objective $\max_{V' \in N(V)} L_a(\Gamma(U'), V')$ (formula (2)), where $V$ is the given test batch, $N(V)$ is a neighborhood around $V$, $L_a$ is a loss function for attack gain, $\Gamma$ is the transductive-learning based defense, and $U' = V'|_X$, the projection of $V'$ to features, is the adversarially perturbed data for breaking $\Gamma$. This

---

Our code is available at: https://github.com/jfc43/eval-transductive-robustness.

[1] We note that this type of defense goes under different names such as "test-time adaptation" or "dynamic defenses". Nevertheless, they all fall into the classic transductive learning paradigm (Vapnik, 1998), which attempts to leverage test data for learning. We thus call them *transductive-learning based defenses*. The word "transductive" is also adopted in Goldwasser et al. (2020).

objective is *transductive* as $U'$, the attacker's output, appears in both attack ($V'$ in $L_a$) and defense ($U'$ in $\Gamma$). We extend this threat model to multiple rounds, which is necessary when considering DENT (Wang et al., 2021) and RMC (Wu et al., 2020b). We point out important subtleties in the modeling that were unclear or overlooked in previous work.

We then study *adaptive attacks*, that is to leverage the knowledge about $\Gamma$ to construct attacks. Compared to situations considered in BPDA (Athalye et al., 2018), a transductive learner $\Gamma$ is even further from being differentiable, and theoretically the attack objective is a bilevel optimization (Colson et al., 2007). To address these difficulties, our key observation is to consider the *transferability of adversarial examples*, and consider a *robust* version of (2): $\max_{U'} \min_{\overline{U} \in \mathcal{N}(U')} L_a(\Gamma(\overline{U}), V')$ (formula (6)), where we want to find a *single* attack set $U'$ to thwart a family of models, induced by $\overline{U}$ "around" $U'$. This objective relaxes the attacker-defender constraint, and provides more information in dealing with nondifferentiability. To solve the robust objective, we propose Greedy Model Space Attack (GMSA), a general attack framework which attempts to solve the robust objective in a greedy manner. GMSA can serve as a new baseline for evaluating transductive-learning based defenses.

We perform a systematic empirical study on various defenses. For RMC (Wu et al., 2020b), DENT (Wang et al., 2021), and URejectron (Goldwasser et al., 2020), we show that even weak instantiations of GMSA can break respective defenses. Specifically, for defenses based on adversarially training, we reduce the robust accuracy to that of adversarial training alone. We note that, under AutoAttack (Croce & Hein, 2020a), the state-of-the-art adaptive attack for the inductive threat model, some of these defenses have claimed to achieve *substantial improvements* compared to *adversarial training alone*. For example, Wang et al. show that DENT can improve the robustness of the state-of-the-art adversarial training defenses by more than 20% absolutely against AutoAttack on CIFAR-10. However, under our adaptive attacks, DENT only has minor improvement: less than 3% improvement over adversarial training alone. Our results thus demonstrates significant differences between attacking transductive-learning based defenses and attacking in the inductive setting, and significant difficulties in the use of transductive learning to improve adversarial robustness. On the positive side, we report a somewhat surprising empirical result of *transductive adversarial training*: Adversarially retraining the model using fresh private randomness on a new batch of test-time data gives a significant increase in robustness against all of our considered attacks.

## 2 RELATED WORK

**Adversarial robustness in the inductive setting.** Many attacks have been proposed to evaluate the adversarial robustness of the defenses in the inductive setting where the model is fixed during the evaluation phase (Goodfellow et al., 2015; Carlini & Wagner, 2017; Kurakin et al., 2017; Moosavi-Dezfooli et al., 2016; Croce & Hein, 2020b). Principles for adaptive attacks have been developed in Tramèr et al. (2020) and many existing defenses are shown to be broken based on attacks developed from these principles (Athalye et al., 2018). A fundamental method to obtain adversarial robustness in this setting is adversarial training (Madry et al., 2018; Zhang et al., 2019). A state-of-the-art attack in the inductive threat model is AutoAttack (Croce & Hein, 2020a).

**Adversarial robustness via test-time defenses**. There have been various work which attempt to improve adversarial robustness by leveraging test-time data. Many of such work attempt to "sanitize" test-time input using a non-differentiable function, and then send it to a pretrained model. Most of these proposals were broken by BPDA (Athalye et al., 2018). To this end, we note that a research agenda for "dynamic model defense" has been proposed in Goodfellow (2019).

**Adversarial robustness using transductive learning.** There has been emerging interesting in using transductive learning to improve adversarial robustness. In view of "dynamic defenses", these proposals attempt to apply transductive learning to the test data and update the model, and then use the updated model to predict on the test data. In this work we consider three such work (Wu et al., 2020b; Wang et al., 2021; Goldwasser et al., 2020).

## 3 PRELIMINARIES

Let $F$ be a model, and for a data point $(\boldsymbol{x}, y) \in \mathcal{X} \times \mathcal{Y}$, a loss function $\ell(F; \boldsymbol{x}, y)$ gives the loss of $F$ on the point. Let $V$ be a set of labeled data points, and let $L(F, V) = \frac{1}{|V|} \sum_{(\boldsymbol{x}, y) \in V} \ell(F; \boldsymbol{x}, y)$ denote

the empirical loss of $F$ on $V$. For example, if we use binary loss $\ell^{0,1}(F; \boldsymbol{x}, y) = \mathbb{1}[F(\boldsymbol{x}) \neq y]$, this gives the test error of $F$ on $V$. We use the notation $V|_X$ to denote the projection of $V$ to its features, that is $\{(\boldsymbol{x}_i, y_i)\}_{i=1}^m|_X \mapsto \{\boldsymbol{x}_i\}_{i=1}^m$. Throughout the paper, we use $N(\cdot)$ to denote a neighborhood function for perturbing features: That is, $N(\boldsymbol{x}) = \{\boldsymbol{x}' \mid d(\boldsymbol{x}', \boldsymbol{x}) < \epsilon\}$ is a set of examples that are close to $\boldsymbol{x}$ in terms of a distance metric $d$ (e.g., $d(\boldsymbol{x}', \boldsymbol{x}) = \|\boldsymbol{x}' - \boldsymbol{x}\|_p$). Given $U = \{\boldsymbol{x}_i\}_{i=1}^m$, let $N(U) = \{\{\boldsymbol{x}_i'\}_{i=1}^m \mid d(\boldsymbol{x}_i', \boldsymbol{x}_i) < \epsilon, i = 0, \ldots, m\}$. Since labels are not changed for adversarial examples, we also use the notation $N(V)$ to denote perturbations of features, with labels fixed.

## 4 Modeling Transductive Robustness

In this section we formulate and analyze threat models for transductive defenses. We first formulate a threat model for a single-round game between the attacker and the defender. We then consider extensions of this threat model to multiple rounds, which are necessary when considering DENT (Wang et al., 2021) and RMC (Wu et al., 2020b), and point out important subtleties in modeling that were not articulated in previous work. We characterize previous test-time defenses using our threat models.

**1-round game**. In this case, the adversary "intercepts" a clean test data $V$ (with clean features $U = V|_X$, and labels $V|_Y$), adversarially perturbs it, and sends a perturbed features $U'$ to the defender. The defender learns a new model based on $U'$. A referee then evaluates the accuracy of the adapted model on $U'$. Formally:

**Definition 1** (*1-round threat model for transductive adversarial robustness*). *Fix an adversarial perturbation type (e.g., $\ell_\infty$ perturbations with perturbation budget $\varepsilon$). Let $P_{X,Y}$ be a data generation distribution. The attacker is an algorithm $\mathcal{A}$, and the defender is a pair of algorithms $(\mathcal{T}, \Gamma)$, where $\mathcal{T}$ is a supervised learning algorithm, and $\Gamma$ is a transductive learning algorithm. A (clean) training set $D$ is sampled i.i.d. from $P_{X,Y}$. A (clean) test set $V$ is sampled i.i.d. from $P_{X,Y}$.*

• *[Training time, defender] The defender trains an optional base model $F = \mathcal{T}(D)$, using the labeled source data $D$.*

• *[Test time, attacker] The attacker receives $V$, and produces an (adversarial) unlabeled dataset $U'$:*

  1. *On input $\Gamma$, $F$, $D$, and $V$, $\mathcal{A}$ perturbs each point $(\boldsymbol{x}, y) \in V$ to $(\boldsymbol{x}', y)$ (subject to the agreed attack type), giving $V' = \mathcal{A}(\Gamma, F, D, V)$ (that is, $V' \in N(V)$).*
  2. *Send $U' = V'|_X$ (the feature vectors of $V'$) to the defender.*

• *[Test time, defender] The defender produces a model as $F^* = \Gamma(F, D, U')$.*

**Multi-round games**. The extension of 1-round games to multi-round contains several important considerations that were implicit or unclear in previous work, and is closely related to what it means by *adaptive attacks*. Specifically:

**Private randomness**. Note that $\Gamma$ uses randomness, such as random initialization and random restarts[2] in adversarial training. Since these randomness are generated *after* the attacker's move, they are treated as *private randomness*, and *not* known to the adversary.

**Intermediate defender states leaking vs. Non-leaking**. In a multi-round game, *the defender* may maintain states across rounds. For example, the defender may store test data and updated models from previous rounds, and use them in a new round. If these intermediate defender states are "leaked" to the attacker, we call it *intermediate defender states leaking*, or simply *states leaking*, otherwise we call it *non states-leaking*, or simply non-leaking. Note that the attacker *cannot* simply compute these information by simulating on the training and testing data, due to the use of *private randomness*. We note that, however, the initial pretrained model is assumed to be known by the attacker. The attacker can also of course maintain arbitrary states, and are assumed not known to the defender.

**Adaptive vs. Non-adaptive**. Because transductive learning happens after the attacker produces $U'$, the attacker may not be able to directly attack the model $\Gamma$ produced. Nevertheless, the attacker is assumed to have *full knowledge of the transductive mechanism $\Gamma$*, except the private randomness. In this paper we call an attack *adaptive* if it makes *explicit use of the knowledge of $\Gamma$*.

**Naturally ordered vs. Adversarially ordered**. Both RMC and DENT handle batches of fixed sizes. An intuitive setup for multi-round game is that the batches come in sequentially, and the attacker

---

[2]When perturbing a data point during adversarial training, one starts with a random point in the neighborhood.

must forward perturbed versions of these batches *in the same order* to the defender, which we call the "naturally ordered" game. However, this formulation does not capture an important scenario: An adversary can wait and pool a large amount of test data, then chooses a *"worst-case" order* of perturbed data points, and then sends them in batches one at a time for adaptation in order to maximize the breach. We call the latter "adversarially ordered" game. We note that all previous work only considered naturally-ordered game, which gives the defender more advantages, and is thus our focus in the rest of the paper. Adversarially-ordered game is evaluated for DENT in Appendix A.7.

**Modeling capacity of our threat models**. Our threat models encompass a large family of defenses. For example, without using $\Gamma$, the threat model degenerates to the classic inductive threat model. Our threat models also capture various "test-time defenses" proposals (e.g., those broken by the BPDA (Athalye et al., 2018)), where $\Gamma$ is a "non-differentiable" function which *"sanitizes" the test data*, instead of updating the model, before sending them to a fixed pretrained model. Therefore, in particular, these proposals are not transductive-learning based. Below we describe previous defenses which we study in the rest of this paper, where $\Gamma$ is indeed transductive learning.

**Example 1** (***Runtime masking and cleansing***). *Runtime masking and cleansing (RMC) (Wu et al., 2020b) is a recent transductive-learning defense. For RMC, the defender is stateful and adapted from the model learned in the last round, on a single test point ($|U| = 1$): The adaptation objective is $F^* = \arg\min_F \sum_{(\boldsymbol{x},y) \in N'(\widehat{\boldsymbol{x}})} L(F,\boldsymbol{x},y)$, where $\widehat{\boldsymbol{x}}$ is the test time feature point, and $N'(\widehat{\boldsymbol{x}})$ is the set of examples in the adversarial training dataset $D'$ that are top-K nearest to $\widehat{\boldsymbol{x}}$ in a distance measure. RMC paper considered two attacks:* **(1) Transfer attack**, *which generates perturbed data by attacking the initial base model, and* **(2) PGD-skip attack**, *which at round $p+1$, runs PGD attack on the model learned at round $p$. In our language, transfer attack is* stateless *(i.e. the adversary maintains no state) and* non-adaptive, *PGD-skip attack is* state-leaking, *but still* non-adaptive.

**Example 2** (***Defensive entropy minimization (DENT (Wang et al., 2021))***). *DENT adapts the model using test input, and can work with any training-time learning procedure. The DENT defender is* stateless: *It always starts the adaptation from the original pretrained model, fixed at the training time. During the test-time adaptation, only the affine parameters in batch normalization layers of the base model are updated, using entropy minimization with the information maximization regularization. In this paper, we show that with strong adaptive attacks under the naturally ordered setting, we are able to reduce the robustness to be almost the same as that of static models (see Section 6). Further, under the adversarially ordered setting, we can completely break DENT.*

**Example 3** (***Goldwasser et al.'s transductive threat model***). *While seemingly our threat model is quite different from the one described in Goldwasser et al. (2020), one can indeed recover their threat model naturally as a 1-round game: First, for the perturbation type, we simply allow arbitrary perturbations in the threat model setup. Second, we have a fixed pretrained model $F$, and the adaptation algorithm $\Gamma$ learns a set $S$ which represents the set of "allowable" points (so $F|_S$ yields a predictor with redaction, namely it outputs $\perp$ for points outside of $S$). Third, we define two error functions as (5) and (6) in Goldwasser et al. (2020):*

$$\operatorname*{err}_{U'}(F|_S, f) \equiv \frac{1}{|U'|} \left| \left\{ \boldsymbol{x}' \in U' \cap S \Big| F(\boldsymbol{x}') \neq f(\boldsymbol{x}') \right\} \right|, \quad \operatorname*{rej}_U(S) \equiv \frac{|U \setminus S|}{|U|} \tag{1}$$

*where $f$ is the ground truth hypothesis. The first equation measures prediction errors in $U'$ that passed through $S$, and the second equation measures the rejection rate of the clean input. The referee evaluates by measuring two errors: $L(F|_S, V') = (\operatorname{err}_{U'}(F|_S), \operatorname{rej}_U(S))$.*

# 5 ADAPTIVE ATTACKS IN ONE ROUND

In this section we study a basic question: *How to perform adaptive attacks against a transductive-learning based defense in* one round? Note that, in each round of a multi-round game, an *independent* batch of test input $U$ is sampled, and the defender can use transductive learning to produce a model specifically adapted to the adversarial input $U'$, *after* the defender receives it. Therefore, it is of fundamental interest to attack this ad-hoc adaptation. We consider white-box attacks: The attacker knows all the details of $\Gamma$, except private randomness, which is sampled after the attacker's move.

We deduce a principle for adaptive attacks in one round, which we call *the principle of attacking model space*: Effective attacks against a transductive defense may need to consider *attacking a set*

*of representative models induced in the neighborhood of $U$*. We give concrete instantiations of this principle, and show in experiments that they break previous transductive-learning based defenses.

**Attacks in multi-round**. If the transductive-learning based defense is stateless, then we simply repeat one-round attack multiple times. If it is stateful, then we need to consider state-leaking setting or non-leaking setting. For all experiments in Section 6, we only evaluate *non-leaking* setting, which is more challenging for the adversary.

## 5.1 GOAL OF THE ATTACKER AND CHALLENGES

To start with, given a defense mechanism $\Gamma$, the objective of the attacker can be formulated as:

$$\max_{V' \in N(V), U' = V'|_X} L_a(\Gamma(F, D, U'), V'). \tag{2}$$

where $L_a$ is the loss function of the attacker. We make some notational simplifications: Since $D$ is a constant, in the following we drop it and write $\Gamma(U')$. Also, since the attacker does not modify the labels in the threat model, we abuse the notation and write the objective as

$$\max_{V', U' = V'|_X} L_a(\Gamma(U'), U'). \tag{3}$$

A generic attacker would proceed iteratively as follows: It starts with the clean test set $V$, and generates a sequence of *(hopefully) increasingly stronger* attack sets $U^{(0)} = V|_X, U^{(1)}, \ldots, U^{(i)}$ ($U^{(i)}$ must satisfy the attack constraints at $U$, such as $\ell_\infty$ bound). We note several basic but important *differences* between transductive attacks and inductive attacks in the classic minimax threat model:

**(D1)** $\Gamma(U')$ **is *not* differentiable**. For the scenarios we are interested in, $\Gamma$ is an optimization algorithm to solve an objective $F^* \in \arg\min_F L_d(F, D, U')$. This renders (3) into a bilevel optimization problem (Colson et al., 2007):

$$\max_{V' \in N(V); U' = V'|_X} L_a(F^*, V') \quad \text{subject to: } F^* \in \arg\min_F L_d(F, D, U'), \tag{4}$$

In these cases, $\Gamma$ is in general *not* (in fact far from) differentiable. A natural attempt is to approximate $\Gamma$ with a differentiable function, using theories such as Neural Tangent Kernels (Jacot et al., 2018). Unfortunately no existing theory applies to the transductive learning, which deals with unlabeled data $U'$ (also, as we have remarked previously, tricks such as BPDA (Athalye et al., 2018) also does not apply because transductive learning is much more complex than test-time defenses considered there).

**(D2)** $U'$ **appears in *both* attack and defense**. Another significant difference is that the attack set $U'$ also appears as the input for the defense (i.e. $\Gamma(U')$). Therefore, while it is easy to find $U'$ to fail $\Gamma(\overline{U})$ for any fixed $\overline{U}$, it is much harder to find a *good direction* to update the attack and converge to *an attack set $U^*$ that fails an entire model space induced by itself:* $\Gamma(U^*)$.

**(D3)** $\Gamma(U')$ **can be a *random variable***. In the classic minimax threat model, the attacker faces a fixed model. However, the output of $\Gamma$ can be a *random variable of models* due to its private randomness, such as the case of Randomized Smoothing (Cohen et al., 2019). In these cases, successfully attacking a single sample of this random variable does not suffice.

---

**Algorithm 1** FIXED POINT ATTACK (FPA)

---

**Require:** A transductive learning algorithm $\Gamma$, an optional training dataset $D$, a clean test set $V$, an initial model $F^{(0)}$, and an integer parameter $T \geq 0$ (the number of iterations).
1: **for** $i = 0, 1, \ldots, T$ **do**
2:     Attack the model obtained in the last iteration to get the perturbed set:

$$V^{(i)} = \arg\max_{V' \in N(V)} L_a(F^{(i)}, V') \tag{5}$$

    where $L_a$ is a loss function. Set $U^{(i)} = V^{(i)}|_X$.
3:     Run the transductive learning algorithm $\Gamma$ to get the next model: $F^{(i+1)} = \Gamma(D, U^{(i)})$.
4: **end for**
5: Select the best attack set $U^{(k)}$ as $k = \arg\max_{0 \leq i \leq T} L(F^{(i+1)}, V^{(i)})$.
6: **return** $U^{(k)}$.

---

**Fixed Point Attack: A first attempt**. We adapt previous literature for solving bilevel optimization in deep learning setting (Lorraine & Duvenaud, 2018) (designed for supervised learning). The idea is simple: At iteration $i + 1$, we fix $U^{(i)}$ and model space $F^{(i)} = \Gamma(U^{(i)})$, and construct $U^{(i+1)}$ to fail it. We call this the Fixed Point Attack (FPA) (Algorithm 1), as one hopes that this process converges to a good fixed point $U^*$. Unfortunately, we found FPA to be weak in experiments. The reason is exactly **(D2)**: $U^{(i+1)}$ failing $F^{(i)}$ may not give any indication that it can also fail $F^{(i+1)}$ induced by itself. Note that transfer attack is a special case of FPA by setting $T = 0$.

## 5.2 Strong adaptive attacks from attacking model spaces

To develop stronger adaptive attacks, we consider a key property of the adversarial attacks: The *transferability of adversarial examples*. Various previous work have identified that adversarial examples transfer (Tramèr et al., 2017; Liu et al., 2016), even across vastly different architectures and models. Therefore, if $U'$ is a good attack set, we would expect that $U'$ also fails $\Gamma(\overline{U})$ for $\overline{U}$ close to $U'$. This leads to the consideration of the following objective:

$$\max_{U'} \min_{\overline{U} \in \mathcal{N}(U')} L_a(\Gamma(\overline{U}), U'). \tag{6}$$

where $\mathcal{N}(\cdot)$ is a neighborhood function (possibly different than $N$). It induces a family of models $\{\Gamma(\overline{U}) \mid \overline{U} \in \mathcal{N}(U')\}$, which we call a *model space*. (in fact, this can be a family of random variables of models) This can be viewed as a natural *robust* version of (3) by considering the transferability of $U'$. While this is seemingly even harder to solve, it has several benefits: **(1) Considering a model space naturally strengthens** FPA**.** FPA naturally falls into this formulation as a weak instantiation where we consider a single $\overline{U} = U^{(i)}$. Also, considering a model space gives the attacker more information in dealing with the nondifferentiability of $\Gamma$ **(D1)**. **(2) It relaxes the attacker-defender constraint (D2).** Perhaps more importantly, for the robust objective, we no longer need the same $U'$ to appear in both defender and attacker. Therefore it gives a natural relaxation which makes attack algorithm design easier.

In summary, while "brittle" $U'$ that does not transfer may indeed exist theoretically, their identification can be challenging algorithmically, and its robust variant provides a natural relaxation considering both algorithmic feasibility and attack strength. This thus leads us to the following principle:

> **The Principle of Attacking Model Spaces**. *An effective adaptive attack against a transductive-learning based defense may need to consider* a model space *induced by a proper neighborhood of* $U$.

---

**Algorithm 2** Greedy Model Space Attack (GMSA)

**Require:** A transductive learning algorithm $\Gamma$, an optional training dataset $D$, a clean test set $V$, an initial model $F^{(0)}$, and an integer parameter $T \geq 0$ (the number of iterations).
1: **for** $i = 0, 1, \ldots, T$ **do**
2:    Attack the previous models to get the perturbed set:

$$V^{(i)} = \arg\max_{V' \in N(V)} L_{\text{GMSA}}(\{F^{(j)}\}_{j=0}^i, V') \tag{7}$$

   where $L_{\text{GMSA}}$ is a loss function. Set $U^{(i)} = V^{(i)} \mid_X$.
3:    Run the transductive learning algorithm $\Gamma$ to get the next model: $F^{(i+1)} = \Gamma(D, U^{(i)})$.
4: **end for**
5: Select the best attack $U^{(k)}$ as $k = \arg\max_{0 \leq i \leq T} L(F^{(i+1)}, V^{(i)})$,
6: **return** $U^{(k)}$.

---

**An instantiation: Greedy Model Space Attack (GMSA)**. We give a simplest possible instantiation of the principle, which we call the *Greedy Model Space Attack* (Algorithm 2). In experiments we use this instantiation to break previous defenses. In this instantiation, the family of model spaces to consider is just all the model spaces constructed in previous iterations. $L_{\text{GMSA}}(\{F^{(j)}\}_{j=0}^i, V')$ is a loss function that the attacker uses to attack the history model spaces. We consider two instantiations: (1) $L_{\text{GMSA}}^{\text{AVG}}(\{F^{(j)}\}_{j=0}^i, V') = \frac{1}{i+1} \sum_{j=0}^i L_a(F^{(j)}, V')$, (2) $L_{\text{GMSA}}^{\text{MIN}}(\{F^{(j)}\}_{j=0}^i, V') = \min_{0 \leq j \leq i} L_a(F^{(j)}, V')$, where $L_{\text{GMSA}}^{\text{AVG}}$ gives attack algorithm GMSA-AVG, and $L_{\text{GMSA}}^{\text{MIN}}$ gives attack algorithm GMSA-MIN. We solve (7) via Projected Gradient Decent (PGD) (the implementation details of GMSA can be found in Appendix A.1.3).

| Dataset | Base Model | Accuracy | | Robustness | | | | | |
|---|---|---|---|---|---|---|---|---|---|
| | | Static | RMC | Static | RMC | | | | |
| | | | | AA | AA | PGD | FPA | GMSA-AVG | GMSA-MIN |
| **MNIST** | Standard | 99.50 | 99.00 | 0.00 | 97.70 | 98.30 | 0.60 | **0.50** | 1.10 |
| | Madry et al. | 99.60 | 97.00 | 87.70 | 95.70 | 96.10 | 59.50 | 61.40 | **58.80** |
| **CIFAR-10** | Standard | 94.30 | 93.10 | 0.00 | 94.20 | 97.60 | 8.50 | **8.00** | 8.10 |
| | Madry et al. | 83.20 | 90.90 | 44.30 | 77.90 | 71.70 | 40.80 | 42.50 | **39.60** |

Table 1: Results of evaluating RMC. We also evaluate the static base model for comparison. **Bold** numbers are worst results.

| Base Model | Accuracy | | Robustness | | | | | | |
|---|---|---|---|---|---|---|---|---|---|
| | Static | DENT | Static | DENT | | | | | |
| | | | AA | DENT-AA | AA | PGD | FPA | GMSA-AVG | GMSA-MIN |
| Wu et al. (2020a) | 85.70 | 86.10 | 58.00 | 78.80 | 64.40 | 59.50 | **59.30** | 59.60 | 59.60 |
| Carmon et al. (2019) | 88.00 | 87.40 | 57.30 | 80.10 | 61.70 | **58.40** | **58.40** | 58.50 | 58.50 |
| Sehwag et al. (2020) | 87.30 | 86.90 | 54.90 | 76.50 | 59.60 | **55.80** | **55.80** | **55.80** | **55.80** |
| Wang et al. (2020) | 86.60 | 85.60 | 53.60 | 75.90 | 61.30 | 55.90 | **55.80** | 56.10 | 56.10 |
| Hendrycks et al. (2019) | 85.80 | 85.50 | 51.80 | 77.20 | 58.40 | **54.20** | 54.40 | **54.20** | **54.20** |
| Wong et al. (2020) | 81.20 | 81.00 | 42.40 | 69.70 | 48.90 | **44.10** | 44.30 | 44.50 | 44.30 |
| Ding et al. (2020) | 82.40 | 82.40 | 39.70 | 62.80 | 44.80 | 39.90 | 39.40 | **39.10** | 39.20 |

Table 2: Results of evaluating DENT on CIFAR-10. We also evaluate the static base model for comparison. **Bold** numbers are worst results.

## 6 EMPIRICAL STUDY

This section evaluates various transductive-learning based defenses. Our main findings are: **(1)** The robustness of existing transductive defenses like RMC and DENT is overestimated. Under our evaluation framework, those defenses either have little robustness or have almost the same robustness as that of the static base model. To this end, we note that while AutoAttack is effective in evaluating the robustness of static models, it is not effective in evaluating the robustness of transductive defenses. In contrast, our GMSA attack is a strong baseline for attacking transductive defenses. **(2)** We experimented a novel idea of applying Domain Adversarial Neural Networks (Ajakan et al., 2014), an unsupervised domain adaptation technique (Wilson & Cook, 2020), as a transductive-learning based defense. We show that DANN has nontrivial and even better robustness compared to existing work, under AutoAttack, PGD attack, and FPA attack, even though it is broken by GMSA. **(3)** We report a somewhat surprising phenomenon on *transductive adversarial training*: Adversarially retraining the model *using fresh private randomness on a new batch of test-time data* gives a significant increase in robustness, against all of our considered attacks. **(4)** Finally, we demonstrated that URejectron, while enjoying theoretical guarantees in the bounded-VC dimensions situation, can be broken in natural deep learning settings.

**Evaluation framework**. For each defense, we report accuracy and robustness. The accuracy is the performance on the clean test inputs, and the robustness is the performance under adversarial attacks. The robustness of transductive defenses is estimated using AutoAttack (AA)[3], PGD attack, FPA, GMSA-MIN and GMSA-AVG. We use PGD attack and AutoAttack in the transfer attack setting for the transductive defense: We generate adversarial examples by attacking a static model (e.g. the base model used by the transductive defense), and then evaluate the transductive defense on the generated adversarial examples. Accuracy and robustness of the static models are also reported for comparison. We always use AutoAttack to estimate the robustness of static models since it is the state-of-the-art for the inductive setting. For all experiments, the defender uses his own private randomness, which is different from the one used by the attacker. Without specified otherwise, all reported values are percentages. Below we give details. Appendix A gives details for replicating the results.

**Runtime Masking and Cleansing (RMC (Wu et al., 2020b))**. RMC adapts the network at test time, and was shown to achieve state-of-the-art robustness under several attacks that are unaware of the defense mechanism (thus these attacks are non-adaptive according to our definition). We follow the setup in Wu et al. (2020b) to perform experiments on MNIST and CIFAR-10 to evaluate the

---

[3]We use the standard version of AutoAttack: https://github.com/fra31/auto-attack/.

| Dataset | Accuracy | | | Robustness | | | | | | |
|---|---|---|---|---|---|---|---|---|---|---|
| | Standard | Madry et al. | DANN | Standard | Madry et al. | DANN | | | | |
| | | | | AA | AA | AA | PGD | FPA | GMSA-AVG | GMSA-MIN |
| **MNIST** | 99.42 | 99.16 | 99.27 | 0.00 | 88.92 | 97.59 | 96.66 | 96.81 | 79.37 | **6.17** |
| **CIFAR-10** | 93.95 | 86.06 | 89.61 | 0.00 | 39.49 | 66.61 | 60.54 | 53.98 | **5.53** | 8.56 |

Table 3: Results of evaluating DANN. **Bold** numbers are worst results.

robustness of RMC. On MNIST, we consider $L_\infty$ norm attack with $\epsilon = 0.3$ and on CIFAR-10, we consider $L_\infty$ norm attack with $\epsilon = 8/255$. The performance of RMC is evaluated on a sequence of test points $\boldsymbol{x}^{(1)}, \cdots, \boldsymbol{x}^{(n)}$ randomly sampled from the test dataset. So we have a $n$-round game. The FPA and GMSA attacks are applied on each round and the initial model $F^{(0)}$ used by the attacks at the $(k+1)$-th round is the adapted model (with calibration in RMC) obtained at the $k$-th round. To save computational cost, we set $n = 1000$. The robustness of RMC is evaluated on a sequence of adversarial examples $\hat{\boldsymbol{x}}^{(1)}, \cdots, \hat{\boldsymbol{x}}^{(n)}$ generated by the attacker on the sequence of test points $\boldsymbol{x}^{(1)}, \cdots, \boldsymbol{x}^{(n)}$. We evaluate the robustness of RMC in the non-state leaking setting with private randomness (both are in favor of the defender).

*Results.* The results are in Table 1. RMC with the standard model is already broken by FPA attack (weaker than GSMA). Compared to the defense-unaware AutoAttack, our GMSA-AVG attack reduces the robustness from $97.70\%$ to $0.50\%$ on MNIST and from $94.20\%$ to $8.00\%$ on CIFAR-10. Further, RMC with adversarially trained model actually provides *worse* adversarial robustness than using adversarial training alone. Under our GMSA-MIN attack, the robustness is reduced from $96.10\%$ to $58.80\%$ on MNIST and from $71.70\%$ to $39.60\%$ on CIFAR-10.

**Defensive Entropy Minimization (DENT (Wang et al., 2021)).** DENT performs test-time adaptation, and works for any training-time learner. It was shown that DENT improves the robustness of the state-of-the-art adversarial training defenses by $20+$ points absolute against AutoAttack on CIFAR-10 under $L_\infty$ norm attack with $\epsilon = 8/255$ (DENT is implemented as a model module, and AutoAttack is directly applied to the module, and we denote this as DENT-AA). Wang et al. also considers adaptive attacks for DENT, such as attacking the static base model using AutoAttack to generate adversarial examples, which is the same as the AutoAttack (AA) in our evaluation.

We evaluate the best version of DENT, called DENT+ in Wang et al., under their original settings on CIFAR-10: DENT is combined with various adversarial training defenses, and only the model adaptation is included without input adaptation. The model is adapted sample-wise for six steps by AdaMod (Ding et al., 2019) with learning rate of 0.006, batch size of 128 and no weight decay. The adaptation objective is entropy minimization with the information maximization regularization. To save computational cost, we only evaluate on 1000 examples randomly sampled from the test dataset. We consider $L_\infty$ norm attack with $\epsilon = 8/255$. We design loss functions for the attacks to generate adversarial examples with high confidence (See Appendix A.3 for the details).

*Results.* Table 2 shows that both DENT-AA and AA overestimate the robustness of DENT. Our PGD attack reduces the robustness of DENT to be almost the same as that of the static defenses. Further, our FPA, GMSA-AVG and GMSA-MIN have similar performance as the PGD attack. The results show that AutoAttack is not effective in evaluating the robustness of transductive defenses.

**Domain Adversarial Neural Network (DANN (Ajakan et al., 2014)).** We consider DANN as a transductive defense for adversarial robustness. We train DANN on the labeled training dataset $D$ (source domain) and unlabeled adversarial test dataset $U'$ (target domain), and then evaluate DANN on $U'$. For each adversarial set $U'$, we train a new DANN model from scratch. We use the standard model trained on $D$ as the base model for DANN. We perform experiments on MNIST and CIFAR-10 to evaluate the adversarial robustness of DANN. On MNIST, we consider $L_\infty$ norm attack with $\epsilon = 0.3$ and on CIFAR-10, we consider $L_\infty$ norm attack with $\epsilon = 8/255$.

*Results.* Table 3 shows that DANN has non-trivial robustness under AutoAttack, PGD attack and FPA attack. However, under our GMSA attack, DANN has little robustness.

**Transductive Adversarial Training (TADV).** We consider a simple but novel transductive-learning based defense called transductive adversarial training: After receiving a set of examples at the test time, we always adversarially retrain the model using fresh randomness. The key point of this transduction is that *private randomness* is sampled after the attacker's move, and so the attacker

| Dataset | Accuracy | | Robustness | | | | | |
| | Madry et al. | TADV | Madry et al. | TADV | | | | |
| | | | AA | AA | PGD | FPA | GMSA-AVG | GMSA-MIN |
|---|---|---|---|---|---|---|---|---|
| **MNIST** | 99.01 | 99.05 | 86.61 | 96.07 | 96.48 | 95.47 | **94.27** | 95.48 |
| **CIFAR-10** | 87.69 | 88.51 | 45.29 | 72.12 | 59.05 | 58.64 | **54.12** | 57.77 |

Table 4: Results of evaluating TADV. **Bold** numbers are worst results.

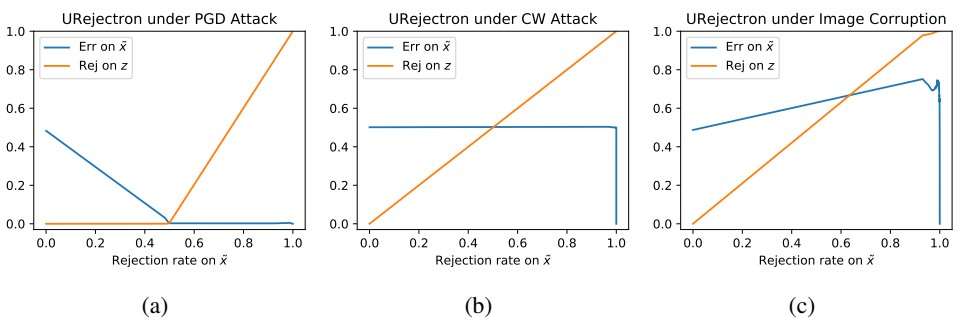

(a)           (b)           (c)

Figure 1: URejectron in three settings. $z$ contains "normal" examples on which the classifier can have high accuracy. $\tilde{x}$ includes $z$ and consists of a mix of 50% "normal" examples and 50% adversarial examples. In (a), the normal examples are clean test inputs and the adversarial examples are generated by PGD attack. In (b), the "normal" examples are still clean test inputs but adversarial examples are generated by CW attack. In (c), the "normal" examples are generated by image corruptions (adversarial examples are generated by PGD attacks).

cannot directly attack the resulting model as in the inductive case. Specifically, for our GMSA attacks, we attack (with loss $L_{\text{GMSA}}^{\text{AVG}}$ or $L_{\text{GMSA}}^{\text{MIN}}$) an ensemble of $T = 10$ models, adversarially trained with independent randomness, and generate a perturbed test set $U'$. Then we adversarially train another model from scratch with independent randomness, and check whether $U'$ transfers to the new model (this thus captures the scenario described earlier). Somewhat surprisingly, we show that $U'$ does not transfer very well, and the TADV improves robustness significantly.

***Results.*** Table 4 shows that transductive adversarial training significantly improves the robustness of adversarial training (Madry et al., 2018). On MNIST, the robustness is improved from 86.61% to 94.27%. On CIFAR-10, the robustness is improved from 45.29% to 54.12%.

**URejectron in deep learning settings**. URejectron performs transductive learning for defense, and has theoretical guarantees under bounded VC dimension case. We evaluated URejectron on GTSRB dataset using ResNet18 network. We used the same implementation by Goldwasser et al..

***Results***. Figure 1(a) shows that for *transfer attacks* generated by PGD attack (Madry et al., 2018), URejectron can indeed work as expected. However, by using different attack algorithms, such as CW attack (Carlini & Wagner, 2017), we observe two failure modes: **(1)** *Imperceptible adversarial perturbations that slip through*. Figure 1(b) shows that one can construct adversarial examples that are very similar to the clean test inputs that can slip through their URejectron construction of $S$ (in the deep learning setting), and cause large errors. **(2)** *Benign perturbations that get rejected*. Figure 1(c) shows that we can generate "benign" perturbed examples using image corruptions, such as slightly increased brightness, but URejectron rejects all.

## 7 CONCLUSION

In this paper, we formulate threat models for transductive defenses and propose an attack framework called Greedy Model Space Attack (GMSA) that can serve as a new baseline for evaluating transductive defenses. We show that GMSA can break previous transductive defenses, which were resilient to previous attacks such as AutoAttack. On the positive side, we show that transductive adversarial training gives a significant increase in robustness against attacks we consider. For the future work, one can explore transductive defenses that can be robust under our GMSA attacks, and can also explore even stronger adaptive attacks that are effective in evaluating transductive defenses.

## ACKNOWLEDGMENTS

The work is partially supported by Air Force Grant FA9550-18-1-0166, the National Science Foundation (NSF) Grants CCF-FMitF-1836978, IIS-2008559, SaTC-Frontiers-1804648, CCF-2046710 and CCF-1652140, and ARO grant number W911NF-17-1-0405. Jiefeng Chen and Somesh Jha are partially supported by the DARPA-GARD problem under agreement number 885000.

## 8  ETHICS STATEMENT

We believe that our work gives positive societal impact in the long term. In the short term potentially some services deploying existing transductive defenses may be broken by adversaries who leverage our new attacks. However, our findings give a necessary step to identify transductive defenses that really work and deepened our understanding of this matter. It also gives positive impact by advancing the science for trustworthy machine learning and potentially how deep transductive learning works.

## 9  REPRODUCIBILITY STATEMENT

We have included enough experimental details to ensure reproducibility in Section 6 and Appendix A. Also, the source code for the experiments is submitted as supplemental materials.

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

# Supplementary Material

## Towards Evaluating the Robustness of Neural Networks Learned by Transduction

## A EXPERIMENTAL DETAILS

### A.1 GENERAL SETUP

#### A.1.1 COMPUTING INFRASTRUCTURE

We run all experiments with PyTorch and NVIDIA GeForce RTX 2080Ti GPUs.

#### A.1.2 DATASET

We use three datasets MNIST, CIFAR-10 and GTSRB in our experiments. The details about these datasets are described below.

**MNIST.** The MNIST (LeCun, 1998) is a large dataset of handwritten digits. Each digit has 5,500 training images and 1,000 test images. Each image is a $28 \times 28$ grayscale. We normalize the range of pixel values to $[0, 1]$.

**CIFAR-10.** The CIFAR-10 (Krizhevsky et al., 2009) is a dataset of 32x32 color images with ten classes, each consisting of 5,000 training images and 1,000 test images. The classes correspond to dogs, frogs, ships, trucks, etc. We normalize the range of pixel values to $[0, 1]$.

**GTSRB.** The German Traffic Sign Recognition Benchmark (GTSRB) (Stallkamp et al., 2012) is a dataset of color images depicting 43 different traffic signs. The images are not of a fixed dimensions and have rich background and varying light conditions as would be expected of photographed images of traffic signs. There are about 34,799 training images, 4,410 validation images and 12,630 test images. We resize each image to $32 \times 32$. The dataset has a large imbalance in the number of sample occurrences across classes. We use data augmentation techniques to enlarge the training data and make the number of samples in each class balanced. We construct a class preserving data augmentation pipeline consisting of rotation, translation, and projection transforms and apply this pipeline to images in the training set until each class contained 10,000 examples. We also preprocess images via image brightness normalization and normalize the range of pixel values to $[0, 1]$.

#### A.1.3 IMPLEMENTATION DETAILS OF THE ATTACKS

We use Projected Gradient Descent (PGD) (Madry et al., 2018) to solve the attack objectives of PGD attack, FPA, GMSA-AVG and GMSA-MIN. For GMSA-AVG, at the $i$-th iteration, when applying PGD on the data point $\boldsymbol{x}$ to generate the perturbation $\delta$, we need to do one backpropagation operation for each model in $\{F^{(j)}\}_{j=0}^{i}$ per PGD step. We do the backpropagation for each model sequentially and then accumulate the gradients to update the perturbation $\delta$ since we might not have enough memory to store all the models and compute the gradients at once, especially when $i$ is large. For GMSA-MIN, we find that it requires more PGD steps to solve the attack objective at the $i$-th iteration where we need to attack $i + 1$ models simultaneously. Thus, we scale the number of PGD steps at the $i$-th iteration by a factor of $i + 1$ for GMSA-MIN.

### A.2 SETUP FOR RMC EXPERIMENTS

We follow the original settings in Wu et al. (2020b) to perform experiments on MNIST and CIFAR-10 datasets to evaluate the adversarial robustness of RMC. We consider two kinds of base models for RMC: one is the model trained via standard supervised training; the other is the model trained using the adversarial training (Madry et al., 2018). We describe the settings for each dataset below.

#### A.2.1 MNIST

**Model architecture and training configuration.** We use a neural network with two convolutional layers, two full connected layers and batch normalization layers. For both standard training and

adversarial training, we train the model for 100 epochs using the Adam optimizer with a batch size of 128 and a learning rate of $10^{-3}$. We use the $L_\infty$ norm PGD attack as the adversary for adversarial training with a perturbation budget $\epsilon$ of 0.3, a step size of 0.01, and number of steps of 40.

**RMC configuration.** We set $K = 1024$. Suppose the clean training set is $\mathbb{D}$. Let $\mathbb{D}'$ contain $|\mathbb{D}|$ clean inputs and $|\mathbb{D}|$ adversarial examples. So $N' = 2|\mathbb{D}|$. We generate the adversarial examples using the $L_\infty$ norm PGD attack with a perturbation budget $\epsilon$ of 0.3, a step size of 0.01, and number of steps of 100. We extract the features from the penultimate layer of the model and use the Euclidean distance in the feature space of the model to find the top-K nearest neighbors of the inputs. When adapting the model, we use Adam as the optimizer and set the learning rate to be $2 \times 10^{-4}$. We train the model until the early-stop condition holds. That is the training epoch reaches 100 or the validation loss doesn't decrease for 5 epochs.

**Attack configuration.** We use the same threat model for all attacks: $L_\infty$ norm perturbation with a perturbation budget $\epsilon$ of 0.3. Cross entropy loss is used as the loss function for PGD attack, FPA, GMSA-AVG and GMSA-MIN: $L_a(F, V) = \frac{1}{|V|} \sum_{(\boldsymbol{x},y) \in V} -\log f(\boldsymbol{x})_y$, where $f(\boldsymbol{x})$ is the softmax output of the model $F$. We use PGD with a step size of 0.01, the number of steps of 100, random start and no restarts. We set $T = 9$ for FPA, GMSA-AVG and GMSA-MIN.

### A.2.2   CIFAR-10

**Model architecture and training configuration.** We use the ResNet-32 network (He et al., 2016). For both standard training and adversarial training, we train the model for 100 epochs using Stochastic Gradient Decent (SGD) optimizer with Nesterov momentum and learning rate schedule. We set momentum 0.9 and $\ell_2$ weight decay with a coefficient of $10^{-4}$. The initial learning rate is 0.1 and it decreases by 0.1 at 50, 75 and 90 epoch respectively. The batch size is 128. We augment the training images using random crop and random horizontal flip. We use the $L_\infty$ norm PGD attack as the adversary for adversarial training with a perturbation budget $\epsilon$ of $\frac{8}{255}$, a step size of $\frac{2}{255}$, and number of steps of 10.

**RMC configuration.** We set $K = 1024$. Suppose the clean training set is $\mathbb{D}$. Let $\mathbb{D}'$ contain $|\mathbb{D}|$ clean inputs and $4|\mathbb{D}|$ adversarial examples. So $N' = 5|\mathbb{D}|$. We generate the adversarial examples using the $L_\infty$ norm PGD attack with a perturbation budget $\epsilon$ of $\frac{8}{255}$, a step size of $\frac{1}{255}$, and number of steps of 40. We extract the features from the penultimate layer of the model and use the Euclidean distance in the feature space of the model to find the top-K nearest neighbors of the inputs. We use Adam as the optimizer and set the learning rate to be $2.5 \times 10^{-5}$.

**Attack configuration.** We use the same threat model for all attacks: $L_\infty$ norm perturbation with a perturbation budget $\epsilon$ of $\frac{8}{255}$. Cross entropy loss is used as the loss function for PGD attack, FPA, GMSA-AVG and GMSA-MIN: $L_a(F, V) = \frac{1}{|V|} \sum_{(\boldsymbol{x},y) \in V} -\log f(\boldsymbol{x})_y$, where $f(\boldsymbol{x})$ is the softmax output of the model $F$. We use PGD with a step size of $\frac{1}{255}$, the number of steps of 40, random start and no restarts. We set $T = 9$ for FPA, GMSA-AVG and GMSA-MIN.

### A.3   SETUP FOR DENT EXPERIMENTS

**DENT configuration.** We perform experiments to evaluate the best version of DENT (DENT+ in Wang et al. (2021)) on CIFAR-10 following the experimental settings in Wang et al. (2021). We use the pre-trained robust models on CIFAR-10 under the $L_\infty$ norm perturbation threat model from RobustBench Model Zoo[4] as the static base models for DENT, including models with model ID Wu2020Adversarial_extra (Wu et al., 2020a), Carmon2019Unlabeled (Carmon et al., 2019), Sehwag2020Hydra (Sehwag et al., 2020), Wang2020Improving (Wang et al., 2020), Hendrycks2019Using (Hendrycks et al., 2019), Wong2020Fast (Wong et al., 2020), and Ding2020MMA (Ding et al., 2020). For the test-time adaptation, only the affine scale $\gamma$ and shift $\beta$ parameters in the batch normalization layers of the base model are updated. DENT updates sample-wise with different affine parameters $(\gamma_i, \beta_i)$ for each input $\boldsymbol{x}_i$. The input adaptation of $\Sigma$ is not used as suggested in Wang et al. (2021). The model is adapted for six steps by AdaMod (Ding et al., 2019) with learning rate of 0.006, batch size of 128 and no weight decay. The adaptation

---

[4]https://github.com/RobustBench/robustbench

objective is entropy minimization with the information maximization regularization:

$$\min_{\theta_i} \sum_{i=1}^{b} -\sum_{c=1}^{C} f(\boldsymbol{x}_i; \theta_i)_c \cdot \log f(\boldsymbol{x}_i; \theta_i)_c + \sum_{c=1}^{C} \sum_{i=1}^{b} f(\boldsymbol{x}_i; \theta_i)_c \cdot \log \sum_{i=1}^{b} f(\boldsymbol{x}_i; \theta_i)_c \qquad (8)$$

where $b$ is the batch size, $C$ is the number of classes and $f(\boldsymbol{x}_i; \theta_i)$ is the softmax output of the model $f$ with the affine parameters $\theta_i = (\gamma_i, \beta_i)$ for the input $\boldsymbol{x}_i$.

**Attack configuration.** We use the same threat model for all attacks: $L_\infty$ norm perturbation with a perturbation budget $\epsilon$ of $\frac{8}{255}$. For PGD attack, FPA, GMSA-AVG and GMSA-MIN, we use the following loss function to find adversarial examples with high confidence: $L_a(F, V) = \frac{1}{|V|} \sum_{(\boldsymbol{x},y) \in V} \max_{k \neq y} f(\boldsymbol{x})_k$, where $f(\boldsymbol{x})$ is the softmax output of the model $F$. However, it is hard to optimize this loss function. Thus, we use two alternative loss functions to find adversarial examples. One is the untargeted CW loss (Carlini & Wagner, 2017): $L_a^1(F, V) = \frac{1}{|V|} \sum_{(\boldsymbol{x},y) \in V} -Z(\boldsymbol{x})_y + \max_{k \neq y} Z(\boldsymbol{x})_k$, where $Z(\boldsymbol{x})$ is the logits of the model $F$ (the output of the layer before the softmax layer). The other is the targeted CW loss: $L_a^2(F, V) = \frac{1}{|V|} \sum_{(\boldsymbol{x},y) \in V} -Z(\boldsymbol{x})_y + Z(\boldsymbol{x})_t$, where $t$ is the targeted label and $t \neq y$. For each attack, we use 14 PGD subroutines to solve its attack objective, including 5 PGD subroutines using the untargeted CW loss $L_a^1$ with different random restarts and 9 PGD subroutines using the targeted CW loss $L_a^2$ with different targeted labels. So for each clean test input $\boldsymbol{x}$, these PGD subroutines will return 14 adversarial examples $\boldsymbol{x}'_1, \ldots, \boldsymbol{x}'_{14}$. Among these adversarial examples, we select the one that maximizes the attack loss with the loss function $L_a(F, V)$ as the final adversarial example $\boldsymbol{x}'$ for $\boldsymbol{x}$. We use the same hyper-parameters for all PGD subroutines: the step size is $\frac{1}{255}$, the number of steps is 100, and the random start is used. We set $T = 2$ for FPA, GMSA-AVG and GMSA-MIN.

## A.4 SETUP FOR DANN EXPERIMENTS

We perform experiments on MNIST and CIFAR-10 datasets. We describe the settings for each dataset below.

### A.4.1 MNIST

**Model architecture.** We use the same model architecture as the one used in Chuang et al. (2020), which is shown below.

| Encoder |
|---|
| nn.Conv2d(3, 64, kernel_size=5) |
| nn.BatchNorm2d |
| nn.MaxPool2d(2) |
| nn.ReLU |
| nn.Conv2d(64, 128, kernel_size=5) |
| nn.BatchNorm2d |
| nn.Dropout2d |
| nn.MaxPool2d(2) |
| nn.ReLU |
| nn.Conv2d(128, 128, kernel_size=3, padding=1) |
| nn.BatchNorm2d |
| nn.ReLU |
| ×2 |

| Predictor |
|---|
| nn.Conv2d(128, 128, kernel_size=3, padding=1) |
| nn.BatchNorm2d |
| nn.ReLU |
| ×3 |
| flatten |
| nn.Linear(2048, 256) |
| nn.BatchNorm1d |
| nn.ReLU |
| nn.Linear(256, 10) |
| nn.Softmax |

| Discriminator |
|---|
| nn.Conv2d(128, 128, kernel_size=3, padding=1) |
| nn.ReLU |
| ×5 |
| Flatten |
| nn.Linear(2048, 256) |
| nn.ReLU |
| nn.Linear(256, 2) |
| nn.Softmax |

**Training configuration.** We train the models for 100 epochs using the Adam optimizer with a batch size of 128 and a learning rate of $10^{-3}$. For the representation matching in DANN, we adopt the original progressive training strategy for the discriminator (Ganin et al., 2016) where the weight $\alpha$ for the domain-invariant loss is initiated at 0 and is gradually changed to 0.1 using the schedule $\alpha = (\frac{2}{1+\exp(-10 \cdot p)} - 1) \cdot 0.1$, where $p$ is the training progress linearly changing from 0 to 1.

**Attack configuration.** We use the same threat model for all attacks: $L_\infty$ norm perturbation with a perturbation budget $\epsilon$ of 0.3. Cross entropy loss is used as the loss function for PGD attack, FPA, GMSA-AVG and GMSA-MIN: $L_a(F, V) = \frac{1}{|V|} \sum_{(\boldsymbol{x},y) \in V} -\log f(\boldsymbol{x})_y$, where $f(\boldsymbol{x})$ is the softmax output of the model $F$. We use PGD with a step size of 0.01, the number of steps of 200, random start and no restarts. We set $T = 9$ for FPA, GMSA-AVG and GMSA-MIN.

### A.4.2 CIFAR-10

**Model architecture.** We use the ResNet-18 network (He et al., 2016) and extract the features from the third basic block for representation matching. The detailed model architecture is shown below.

| Encoder |
|---|
| nn.Conv2d(3, 64, kernel_size=3) |
| nn.BatchNorm2d |
| nn.ReLU |
| BasicBlock(in_planes=64, planes=2, stride=1) |
| BasicBlock(in_planes=128, planes=2, stride=2) |
| BasicBlock(in_planes=256, planes=2, stride=2) |

| Predictor |
|---|
| BasicBlock(in_planes=512, planes=2, stride=2) |
| avg_pool2d |
| flatten |
| nn.Linear(512, 10) |
| nn.Softmax |

| Discriminator |
|---|
| BasicBlock(in_planes=512, planes=2, stride=2) |
| avg_pool2d |
| flatten |
| nn.Linear(512, 2) |
| nn.Softmax |

**Training configuration.** We train the models for 100 epochs using stochastic gradient decent (SGD) optimizer with Nesterov momentum and learning rate schedule. We set momentum 0.9 and $\ell_2$ weight decay with a coefficient of $10^{-4}$. The initial learning rate is 0.1 and it decreases by 0.1 at 50, 75 and 90 epoch respectively. The batch size is 64. We augment the training images using random crop and random horizontal flip. For the representation matching in DANN, we adopt the original progressive training strategy for the discriminator (Ganin et al., 2016) where the weight $\alpha$ for the domain-invariant loss is initiated at 0 and is gradually changed to 1 using the schedule $\alpha = \frac{2}{1+\exp(-10 \cdot p)} - 1$, where $p$ is the training progress linearly changing from 0 to 1.

**Attack configuration.** We use the same threat model for all attacks: $L_\infty$ norm perturbation with a perturbation budget $\epsilon$ of $\frac{8}{255}$. Cross entropy loss is used as the loss function for PGD attack, FPA, GMSA-AVG and GMSA-MIN: $L_a(F, V) = \frac{1}{|V|} \sum_{(\boldsymbol{x},y) \in V} -\log f(\boldsymbol{x})_y$, where $f(\boldsymbol{x})$ is the softmax output of the model $F$. We use PGD with a step size of $\frac{1}{255}$, the number of steps of 100, random start and no restarts. We set $T = 9$ for FPA, GMSA-AVG and GMSA-MIN.

## A.5 SETUP FOR TADV EXPERIMENTS

We perform experiments on MNIST and CIFAR-10 datasets. We describe the settings for each dataset below.

### A.5.1 MNIST

**Model architecture and Training configuration.** We use the LeNet network architecture. We train the models for 100 epochs using the Adam optimizer with a batch size of 128 and a learning rate of $10^{-3}$. We use the $L_\infty$ norm PGD attack as the adversary to generate adversarial training examples with a perturbation budget $\epsilon$ of 0.3, a step size of 0.01, and number of steps of 40. We train on 50% clean and 50% adversarial examples per batch.

**Attack configuration.** We use the same threat model for all attacks: $L_\infty$ norm perturbation with a perturbation budget $\epsilon$ of 0.3. Cross entropy loss is used as the loss function for PGD attack, FPA, GMSA-AVG and GMSA-MIN: $L_a(F, V) = \frac{1}{|V|} \sum_{(\boldsymbol{x},y) \in V} - \log f(\boldsymbol{x})_y$, where $f(\boldsymbol{x})$ is the softmax output of the model $F$. We use PGD with a step size of 0.01, the number of steps of 200, random start and no restarts. We set $T = 9$ for FPA, GMSA-AVG and GMSA-MIN.

### A.5.2 CIFAR-10

**Model architecture and Training configuration.** We use the ResNet-20 network architecture (He et al., 2016). We train the models for 110 epochs using stochastic gradient decent (SGD) optimizer with Nesterov momentum and learning rate schedule. We set momentum 0.9 and $\ell_2$ weight decay with a coefficient of $5 \times 10^{-4}$. The initial learning rate is 0.1 and it decreases by 0.1 at 100 and 105 epoch respectively. The batch size is 128. We augment the training images using random crop and random horizontal flip. We use the $L_\infty$ norm PGD attack as the adversary to generate adversarial training examples with a perturbation budget $\epsilon$ of $\frac{8}{255}$, a step size of $\frac{2}{255}$, and number of steps of 10. We train on 50% clean and 50% adversarial examples per batch.

**Attack configuration.** We use the same threat model for all attacks: $L_\infty$ norm perturbation with a perturbation budget $\epsilon$ of $\frac{8}{255}$. Cross entropy loss is used as the loss function for PGD attack, FPA, GMSA-AVG and GMSA-MIN: $L_a(F, V) = \frac{1}{|V|} \sum_{(\boldsymbol{x},y) \in V} - \log f(\boldsymbol{x})_y$, where $f(\boldsymbol{x})$ is the softmax output of the model $F$. We use PGD with a step size of $\frac{1}{255}$, the number of steps of 100, random start and no restarts. We set $T = 9$ for FPA, GMSA-AVG and GMSA-MIN.

## A.6 SETUP FOR UREJECTRON EXPERIMENTS

We use a subset of the GTSRB augmented training data for our experiments, which has 10 classes and contains 10,000 images for each class. We implement URejectron (Goldwasser et al., 2020) on this dataset using the ResNet18 network (He et al., 2016) in the transductive setting. Following Goldwasser et al. (2020), we implement the basic form of the URejectron algorithm, with $T = 1$ iteration. That is we train a discriminator $h$ to distinguish between examples from $P$ and $Q$, and train a classifier $F$ on $P$. Specifically, we randomly split the data into a training set $D_{\text{train}}$ containing 63,000 images, a validation set $D_{\text{val}}$ containing 7,000 images and a test set $D_{\text{test}}$ containing 30,000 images. We then use the training set $D_{\text{train}}$ to train a classifier $F$ using the ResNet18 network. We train the classifier $F$ for 10 epochs using Adam optimizer with a batch size of 128 and a learning rate of $10^{-3}$. The accuracy of the classifier on the training set $D_{\text{train}}$ is 99.90% and its accuracy on the validation set $D_{\text{val}}$ is 99.63%. We construct a set $\tilde{x}$ consisting of 50% normal examples and 50% adversarial examples. The normal examples in the set $\tilde{x}$ form a set $z$. We train the discriminator $h$ on the set $D_{\text{train}}$ (with label 0) and the set $\tilde{x}$ (with label 1). We then evaluate URejectron's performance on $\tilde{x}$: under a certain threshold used by the discriminator $h$, we measure the fraction of normal examples in $z$ that are rejected by the discriminator $h$ and the error rate of the classifier $F$ on the examples in the set $\tilde{x}$ that are accepted by the discriminator $h$. The set $z$ can be $D_{\text{test}}$ or a set of corrupted images generated on $D_{\text{test}}$. We use the method proposed in Hendrycks & Dietterich (2019) to generate corrupted images with the corruption type of brightness and the severity level of 1. The accuracy of the classifier on the corrupted images is 98.90%. The adversarial examples in $\tilde{x}$ are generated by the PGD attack (Madry et al., 2018) or the CW attack (Carlini & Wagner, 2017). For PGD attack, we use $L_\infty$ norm with perturbation budget $\epsilon = \frac{8}{255}$ and random initialization. The number of iterations is 40 and the step

| Base Model | Robustness | | | | | |
|---|---|---|---|---|---|---|
| | Static | DENT | | | | |
| | AA | AA | PGD | FPA | GMSA-AVG | GMSA-MIN |
| Wu et al. (2020a) | 58.00 | 58.00 | 50.40 | **50.30** | 50.40 | 50.40 |
| Carmon et al. (2019) | 57.30 | 58.00 | **51.80** | **51.80** | **51.80** | **51.80** |
| Sehwag et al. (2020) | 54.90 | 55.90 | 50.10 | **49.80** | 50.00 | 50.00 |
| Wang et al. (2020) | 53.60 | 55.90 | 49.00 | 49.10 | 48.90 | **48.70** |
| Hendrycks et al. (2019) | 51.80 | 53.10 | **48.10** | 48.20 | 48.30 | 48.30 |
| Wong et al. (2020) | 42.40 | 44.60 | 40.10 | **39.90** | 40.00 | 40.00 |
| Ding et al. (2020) | 39.70 | 42.10 | 36.20 | 35.70 | 35.30 | **35.00** |

Table 5: Results of evaluating DENT on CIFAR-10 under the adversarially-ordered game. **Bold** numbers are worst results.

| Dataset | Base Model | Accuracy | | Robustness | | |
|---|---|---|---|---|---|---|
| | | Static | RMC | Static | RMC | |
| | | | | AA | GMSA-AVG | GMSA-MIN |
| **MNIST** | Standard | 99.40±0.15 | 98.62±0.23 | 0.00±0.00 | 0.54±0.05 | 0.80±0.19 |
| | Madry et al. | 99.24±0.22 | 96.50±0.91 | 87.86±0.71 | 57.14±5.83 | 59.48±4.52 |
| **CIFAR-10** | Standard | 93.96±0.38 | 93.56±0.34 | 0.00±0.00 | 8.50±1.29 | 8.92±0.93 |
| | Madry et al. | 83.70±1.11 | 91.46±0.71 | 43.58±1.30 | 38.50±2.07 | 39.00±1.58 |

Table 6: Results of evaluating RMC. We also evaluate the static base model for comparison. We report the mean and standard deviation of the accuracy or robustness (mean±std) over the five random runs of the experiment.

size is $\frac{1}{255}$. The robustness of the classifier under the PGD attack is 3.66%. For CW attack, we use $L_2$ norm as distance measure and set $c = 1$ and $\kappa = 0$. The learning rate is 0.01 and the number of steps is 100. The robustness of the classifier under the CW attack is 0.00%.

## A.7 EVALUATE DENT UNDER THE ADVERSARIALLY-ORDERED GAME

We evaluate the robustness of DENT under the adversarially-ordered game where the adversary can choose a "worst-case" order of perturbed data points after receiving a large amount of test data and then sends them in batches one at a time to the defender. Specifically, each time the attacker will generate adversarial examples on up to 256 data points, and then sort the adversarial examples by their labels from lowest to highest, and finally send the sorted adversarial examples in batches one at a time to the defender. Other experimental settings are the same as those described in Appendix A.3. The results in Table 5 show that under the adversarially-ordered game, we can reduce the robustness of DENT to be lower than that of static base models.

## A.8 MULTIPLE RANDOM RUNS OF THE RMC EXPERIMENT

In Section 6, we describe the experimental setup for evaluating RMC. The performance of RMC is evaluated on a sequence of test points $\boldsymbol{x}^{(1)}, \cdots, \boldsymbol{x}^{(n)}$ randomly sampled from the test dataset. We repeat this experiment five times with different random seeds and report the mean and standard deviation of the results over the multiple random runs of the experiment. When evaluating the robustness of RMC, we only use the GMSA-AVG attack and the GMSA-MIN attack since they are the strongest attacks. From Table 6, we can see that the results don't vary much across different random runs and the conclusion that the proposed GMSA attacks can break RMC still holds.

