# OpenReview forum: "Towards Evaluating the Robustness of Neural Networks Learned by Transduction"
_ICLR.cc/2022/Conference — ICLR 2022 Poster_

### Official Review · Reviewer_WKKJ · 2021-11-02

**Correctness:** 3
**Technical Novelty And Significance:** 3
**Empirical Novelty And Significance:** 3
**Recommendation:** 5
**Confidence:** 5

**Main Review:**

My biggest concern is about the importance of the studied topic in this paper. Specifically, the so-called transductive defenses are not very popular and have not been fully recognized by the community so far. In fact, I happen to have read the paper of RMC when it was published in ICML and thought that the authors of RMC should have tested some adaptive defense-aware attacks in their work, which they did not. After all, breaking random or dynamic models for attacking purpose has been investigated by BPDA and EOT. Meanwhile, I also served as a reviewer for Dent, which was rejected at the time and has not been accepted by other conferences according to my knowledge. In the review opinions for Dent, many reviewers also questioned that it failed to conduct some defense-aware and adaptive evaluations. Thus, to me, it seems that the proposed attack GMSA origins from the obfuscated gradient work and only breaks a few defenses that are somewhat not canonical.

**Summary Of The Paper:**

This paper intends to break a specific type of adversarial defenses. These defenses choose to dynamically modify the model for the purpose of robustness and defenses. This paper proposes to counter such defenses with a new optimization algorithm.

**Summary Of The Review:**

I give a weak reject score for now. I may change my score after I exchange my opinions with other reviewers.

---

> ### Author Response · Authors · 2021-11-11
> **Thanks for the comments and some thoughts on your concerns**
>
> We thank the reviewer for the interesting questions challenging the importance of this work. We agree that transductive-learning based defenses are currently far from being popular, and they look very similar to the old and failed idea of "test-time defenses", since the line of work by Carlini and others seems to have "killed" test-time defenses altogether. On the other hand:
>
> 1. Transductive-learning based defenses seem to be genuinely different. Theoretically, the work by Goldwasser, Kalai, Kalai, and Montasser has given provable evidence that transductive learning gives surprisingly new guarantees in the bounded VC regime, also, we note that just a few weeks early, Montasser, Hanneke, and Srebro (https://arxiv.org/abs/2110.10602) have given another theory paper which shows very interesting theory results about transductive robustness. None of these appeared in the old "test-time defenses" line.
>
> 2. The real distinction is this: Traditional test-time defenses are still too "inductive", you train a large model and fix it, and the test-time defense part is really kind of some very simplistic procedure that does "input sanitization". This is actually the core reason why attacks like BPDA can still work well against such test-time defenses.
>
> Transductive-learning based defenses seem to be genuinely different also in practice – There is an actual learning algorithm that tries to leverage the unlabeled data to update the model. For example, in our study, we used Domain Adversarial Neural Networks (DANN) training as a transductive learning based defense, and it is far beyond what "test-time defenses" was about, and using DANN, we train models completely from scratch, without even a pretrained model to start with.
>
> For these defenses, it is really unclear at all how BPDA could possibly be applied, and it is not until we discovered "model space attacks" described in Section 5, and its simple instantiation of GMSA, that we can break these defenses. To this end, the "transductive adversarial training" we give later in the paper gives surprising results that **fresh and private randomness to adversarially train the model from scratch gives a significant increase in robustness**. This is novel and intriguing.
>
> As another note, one may question whether it is a practical assumption of having a large batch for the transductive learning to be applicable – this is in fact very practical – many practical ML pipelines are deployed in the batch mode (e.g., https://cloud.google.com/ai-platform/prediction/docs/batch-predict), where the pipeline waits until a large batch of test inputs is collected and then makes predictions. There is an opportunity to adapt the model based on these test-time inputs using transductive learning.
>
> 3. We also observe that there is a gap between the theory and practice in transductive-learning based defenses. The theory suggests that there is something real in this direction, but in the practical DL setting, there are only several instances. We believe that given the theory progress, there will be more and more practical work to explore whether we can do something new in this direction, and in fact, transductive adversarial training is discovered by this exploration already in this paper.
>
> For any of these future explorations to happen, the first thing to establish is a rigorous framework for evaluation – which is what we want to achieve in this paper, and we believe that we have made a solid contribution for this purpose.

---

### Official Review · Reviewer_q4Dr · 2021-11-02

**Correctness:** 4
**Technical Novelty And Significance:** 3
**Empirical Novelty And Significance:** 3
**Recommendation:** 6
**Confidence:** 4

**Main Review:**

Strengths:
- I appreciate the formalism of the adversarial game for transductive learning introduced by the paper, which also seems novel.
- The paper covers a wide range of topics: formalizing the transductive attacks and defenses, breaking transductive defenses thought to be strong, experimenting with domain adaptation as defense against the attack proposed in the paper.
- The proposed attack seems to perform well, defeating existing defenses. The experimental section seems to follow standard practices for robustness evaluation.
- The implementation is provided.
- The paper is well-written.

Weaknesses:
- The topic of the computation cost of the proposed attack is not discussed.
- The proposed method seems to be a variation of PGD (limited novelty), but considering that the paper has other contributions and GMSA works well in practice, this work is arguably worth publishing.

Other comments / questions:
- The paper could benefit from additional proofreading.
- The transferred adversarial samples produced with PGD and AutoAttack might be weaker than some generated with the proposed attack, thus not resulting in a proper evaluation.

**Summary Of The Paper:**

This paper proposes a new attack strategy for adversarial examples in the case where transductive defense strategies are employed. The proposed attack is based on projected gradient descent and manages to break existing transductive defenses that were thought to perform well. A new strategy for adversarial training is also proposed yielding more robust models in the transductive setting than existing defenses. Experiments are performed on MNIST, CIFAR-10 and GTSRB, against both naturally trained and robust models.

**Summary Of The Review:**

Good formalism for adversarial machine learning in transductive setting, plus other smaller contributions.

---

> ### Author Response · Authors · 2021-11-11
> **Thanks for the encouraging comments and on GMSA**
>
> We thank the reviewer for the encouraging comments. We will revise the writing and add discussion for computation costs.
>
> Regarding the PGD comment (which we believe is a main point): We agree that eventually it boils down to something somewhat along the line of "PGD attack against an ensemble" (note that GMSA is about attacking **one batch of test input, namely attack for one-round**), however the non-trivial point is **what ensemble we should consider**. Given one batch of clean input $U$, our analysis (in Section 4) indicates that one should consider the ensemble of models induced by $U'$ in the neighborhood $U$, which we call the **model space** (for details see Section 5.2). We believe that the concept of model space will be useful for all future works that want to evaluate a transductive-learning-based defense.

---

### Official Review · Reviewer_7duz · 2021-11-04

**Correctness:** 4
**Technical Novelty And Significance:** 2
**Empirical Novelty And Significance:** 3
**Recommendation:** 6
**Confidence:** 4

**Main Review:**

**Strengths**
- The paper is generally well-written and clear. It clearly articulates issues with the previous evaluation of transductive learning algorithms for learning and lays out the threat model that should be used in the future when evaluating these algorithms. It also delves into the various aspects of the multi-round game that results from an attack on transductive robust learning.
- The empirical study is fairly thorough, tackling 4 different types of robust learning algorithms across two standard datasets, using the two different attacks proposed in the paper.
**Weaknesses**
- While the paper does a good job of articulating the challenges for an adaptive attacker on transductive algorithms, including that the problem results in a bilevel optimization problem for the attacker, it does not use particularly novel methods to tackle this problem. Both proposed methods just turn the problem into a static optimization problem which just use PGD to solve it. There is a plethora of literature on end-to-end poisoning attacks (see [1], [2]) that uses methods such as gradient unrolling or replacing the model with an approximation derived from the Neural Tangent Kernel. The GSMA attack that is proposed merely uses the history of all the trained models, and is essentially still a static attack. I would urge the authors to understand the methods used in the poisoning literature for circumventing the challenges of bilevel optimization.
- Some aspects of the manner in which the final robustness numbers are determined for the proposed attacks are unclear. Specifically, when the transductive model is run for T iterations, is the robustness reported using the best adversarial example set over all iterations, but evaluated with respect to the final model with all test data? Further, for DENT, the value of T is set to 2, so does that mean the numbers reported are only with respect to two samples from the test set? I read through the paper and appendix multiple times for clarity on this point, but it did not seem to be addressed.
- The strawman TADV approach is not evaluated with sufficient rigor. The private randomness can be circumvented by the attacker simulating adversarial training themselves with multiple instantiations of the initialization and adversarial example generation, and then generating universal adversarial perturbations to be effective across all these locally generated models. It would be interesting to see how the defense performs under that setting.

**References**
[1] Towards Poisoning of Deep Learning Algorithms with Back-gradient Optimization, https://arxiv.org/abs/1708.08689
[2] Neural Tangent Generalization Attacks , https://proceedings.mlr.press/v139/yuan21b.html


**Summary Of The Paper:**

This paper proposes adaptive attacks against transductive robust learning methods. These methods aim to improve the robustness of models to adversarial examples by updating the model using unlabeled test data. However, this paper notes that previous evaluation of transductive robust learning has not considered attacks that are truly adaptive, i.e. the attacker should be able to utilize its knowledge of the transductive learning algorithm to craft better examples. Two types of adaptive attacks against transductive learning algorithms are proposed, a single step method and a greedy method taking into account all previous instances of the model learned during transductive robust learning. The experimental results on two existing training methods and two newly proposed ones, one of which is a strawman, demonstrate that the proposed adaptive attacks render transductive robust training to be no more robust than standard robust training. The paper's conclusion is that the previously claimed improvements in robustness were an artifact of the use of non-adaptive attacks.

**Summary Of The Review:**

Overall, while I found the paper to elucidate an important shortcoming in previous work on evaluating the robustness of transductive robust learning, the attack methods used lacked novelty. In fact, in a number of cases the simple FPA algorithm had the best performance. I urge the authors to consider improved methods to circumvent the bilevel optimization problem.

+++++++++++++++++++

The clarifications have improved the paper. While I still believe more innovation is possible for the attack, I have decided to raise my score to a 6 since the paper is likely to lead to interesting follow-up work.

---

> ### Author Response · Authors · 2021-11-11
> **Thanks for the comments and clarifications**
>
> We thank the reviewer for the detailed comments. Below we respond to the bullets you listed. We hope that it will clarify, and we will revise the writing accordingly (and the writing revision is straightforward according to our understanding). Please feel free to let us know your further questions.
>
> **Your second bullet**. Our team has quite a discussion about this bullet. We believe that you may have some confusions: The attack of GMSA, and in fact, the entire Section 5, is about **how to attack** in **one round**, namely to generate **one batch** of perturbed input in order to fool the defender – and you may have confused GMSA as for attacking multiple rounds (where multiple batches are generated), and those models in formula (7) of the GMSA algorithm as "historical models". This is not the case.
>
> For GMSA, what we consider is that in one round, given one batch of clean input $U$, the attacker tries to generate multiple models in the neighborhood of $U$ (see (7), each time we pick a $V'$ in the neighborhood of $V$), and then attacks them all in order to generate one batch of perturbed input $U'$ to send to the defender.
>
> So our attack is not static, it tries to construct the "model space" of $\Gamma$ in the neighborhood of $U$ and tries to break them all – that's the key point of model space attack against a transductive learning algorithm $\Gamma$.
>
> We realized that our writing is not clear that Section 5 is about how to attack in one round. We apologize for that and will revise the writing. Basically, currently for attacking multiple rounds, we just invoke GMSA multiple times.
>
> **Your third bullet**. That's exactly what we did in the evaluation of TADV using GMSA attack – The attacker adversarially trains 10 models using different randomness(note that adversarial training has randomness beyond the random initialization of the model) and tries to attack all the resulting models (of course the attacker does not know the final randomness used by the defender, which happens after the attacker sends the perturbed data to the defender). We have also tried to increase it to 15, and there is no further improvement on the effectiveness of the attack – 15 has been approaching our current computational limit and so we stopped there. We will clarify this in the writing, which is easily doable.
>
> **Your first bullet**. Now let's go back to your first bullet. First, as we mentioned earlier, GMSA is not static against one model. Second, we are very familiar with the line of poisoning attacks as well as the theory of NTK (in fact FPA is an adaptation of previous work, abs/1802.09419, in addressing bilevel optimization in the DL setting). The question is that it is really unclear how those techniques can be applied in our setting (and in fact GMSA is an attempt to adapt). To this end for example, NTK works in the case where it is a supervised learning with labeled data, but $\Gamma$ is transductive – for example it could be Domain Adversarial Neural Networks – that makes heavy use of the unlabeled data – and the optimization strategy is somewhat a GAN-style training – and it's even not clear what an "NTK theory" would be for such scenarios.

---

### Official Review · Reviewer_2Lpz · 2021-11-08

**Correctness:** 3
**Technical Novelty And Significance:** 2
**Empirical Novelty And Significance:** 2
**Recommendation:** 5
**Confidence:** 4

**Main Review:**

## Strength
1. Experimental results seem to suggest the proposed GMSA is effective (though there are some places not clear and require clarification, see below questions)
2. The proposed idea (greedy model space attack) is simple yet seemingly effective

## Weakness
1. The writing needs to be improved. The current draft is written in a complicated way, which is hard to parse, e.g. the notations, the problem setting explanation etc.
2. The are some questions/places not clear and require clarifications from the authors, see below.

### Questions
Q1: How practical is the problem setting of adversarial robustness in transductive learning? Also, it's a bit not clear about how exactly the adversary and the defender interact. I assume it works as follows:
adversary generate a batch of adv examples (adv_0) based on model_0 (original model) --> defender get adv_0 and update the model to model_1 --> adversary generate a new batch of adv example (adv_1) based on model_1 --> defender get adv_1 and update the model_1 to model_2 --> repeat to the end

- Are my above understanding correct? If so, then is it practical to assume that the adversary and defender will interact in this way?

- Perhaps a more difficult setting is when the adversary have no ideas of when the defender actually update the model? In this case, would the proposed method work (it seems that the proposed method is a white-box attack setting)?

Q2: it's a bit not clear to me regarding the "private randomness".
- What happens if the adversary also know about the randomness in $\Gamma$? The attacker should be able to develop a stronger attack?
- The authors mention that the $\Gamma$ can be randomized smoothing type of defense, but I didn't see the experiment results compared with randomized smoothing? Can the authors show some experiments for this?

Q3: Table 1, Table 2
- Table 1: It's not clear to me how exactly is the difference between the AA under static and AA under RMC? Using my understanding in Q1, does it mean that, AA (static) refers to attacking model_0, while AA (RMC) refers to attack model_i, while i = {1, 2, 3, ...} ? Then why AA (RMC) does not work?
- Table 2: can the authors explain what's the difference between DENT-AA and AA under DENT? why they have a large gap (~20%) in robust accuracy?
- Can the authors explain again what's the difference between DENT and RMC? It looks like the proposed GSMA is more effective on RMC compared to AA (RMC) while not so effective compared to AA (DENT)?

Q4: the authors mention the setting of adversarially ordered and naturally ordered data. But from the Algorithm 1 and 2, the data order doesn't seem to matter in GMSA? Why DENT and RMC need the naturally ordered data?



**Summary Of The Paper:**

This paper study the adversarial robustness in transductive learning, where the defenders can update the models during test time by interacting with the adversary in multiple rounds. The authors propose "Greedy Model Space Attack" (GSMA), where the key idea is to general adversarial examples considering all the history models in the past rounds. The result show that the proposed attack can decrease robust accuracy more than other baseline (the static AA, DENT AA).

**Summary Of The Review:**

In summary, this paper propose a simple yet effective GSMA attack to evaluate adversarial robustness of models in the transductive learning setting. Some results are promising but there are a few places require further explanation/clarification.

---

> ### Author Response · Authors · 2021-11-11
> **Thanks for the comments and clarifications for Q1 and Q2**
>
> We thank the reviewer for the detailed comments. Below we answer your questions and we hope that clarifies.
>
> **Q1**. The scenario we model after is very practical: Production ML pipelines (e.g. Google) typically are deployed in batch mode, namely it waits until a large batch of test input is collected (these inputs can be adversarially perturbed, which is what we consider here), and we are modeling the fact that these pipelines can adapt the ML model based on these inputs, before making predictions on the inputs. This is the transductive-learning based defense this paper is about. The same scenarios have also been considered in RMC, DENT, and the theoretical work by Goldwasser et al., and another very recent theory work by Montasser, Hanneke, and Srebro (https://arxiv.org/abs/2110.10602). Our modeling generalizes all these works (for example, we are the only work that has rigorously formalized the multi-round case).
>
> For the process you described, we assume that you are talking about the **states-leaking** case in a multi-round game. If so, then yes you are right, every time after the defender updates the model, "states leaking" means that the updated model is leaked to the adversary. However, for all defenses evaluated, we break them in the **no-state leaking** scenario; so our attacks indeed make no "state-leaking" assumptions. Now more details:
>
> 1. The process you mentioned misses a key point why transductive learning matters here: Think about the first round: The adversary sends a batch of perturbed input $U_0'$ to the defender, the defender updates the model based on $U_0'$ to get $M_1$, and make predictions on $U_0'$ – note here that this accuracy is evaluated on the same $U_0'$ the adversary sends in, and even though this updated model is leaked to the adversary (in the state-leaking case) it will only benefit the adversary to generate the **next** batch of perturbed input, not the first batch $U_0'$. Similarly, of course the adversary can attack $M_1$ to generate the second batch of perturbed input $U_1'$, but no matter what the adversary does – the defender will update the model based on $U_1'$, namely the defender takes **the last move** before making predictions. This is the key for test-time adaptation/transductive-learning based defenses. We hope this is clear now to the reviewer.
>
> 2. For all the defenses we evaluated in this paper, we break them in **no-state leaking** case (this is mentioned in Section 6, for example, when we described RMC; see the last sentence).
>
> 3. The "states leaking" scenario is commonly studied in security research. Note that either we study no-state leaking where absolutely no state is leaked at all, or we assume that every state is leaked. It is unclear how to model that sometimes it is leaked and sometimes it is not. To this end, we list such things here because in the RMC paper they do consider an attack in the "states-leaking" case, but that attack is even weaker than our "no-state leaking" attack.
>
> **Q2**. In our case, private randomness is sampled by the defender during the "test-time adaptation" (or transductive learning phase), which happens **after** the adversary sends the perturbed test-time input to the defender, so to the attacker the "private randomness" happens in the **future** and is unknown to him when he prepares the perturbed input for the defender.
>
> But to your question – if the attacker sees the future and magically knows the randomness used by the defender during test-time adaptation, then the attacker can of course achieve stronger attacks. In fact, this is exactly why transductive adversarial training has superior robustness to inductive adversarial training: For the former, the randomness used by the defender is unknown to the adversary (because the defender can train a model **from scratch** using fresh randomness, **after** receiving the test-time input from the attacker), while in the latter, since the attacker knows all the randomness, the attacker can directly attack the model as a result of the training.
>
> Regarding your second bullet. Our threat model does allow many different $\Gamma$, including Randomized Smoothing. However, as we have mentioned in the paragraph **Modeling capacity of our threat models** (pp. 4), many of these $\Gamma$ are **not** actually transductive learning (there is no actual learning for updating the model). Randomized Smoothing is a test-time defense, but there is **no** learning in it or the model is not updated, so it is not a test-time defense via transductive learning. For this reason, we believe it is beyond scope to evaluate Randomized Smoothing.

---

> > ### Author Response · Authors · 2021-11-11
> > **Thanks for the comments and clarifications for Q3 and Q4**
> >
> > **Q3**. For the first bullet, AA (static) is the robustness of the **static model** for comparison (static model is the traditional inductive case, where the model is trained and fixed during the test time, and the attacker is assumed to know the model and just attack it using PGD say).
> >
> > Regarding your description of the AA for RMC –it seems to us that this may be related to the potential confusion about how transductive-learning based defense works. For the first round, the updated model $M_1$ is generated **after** the adversary prepared the first batch of perturbed input, because $M_1$ is generated by running a transductive learning algorithm on the first batch of perturbed input! The model $M_1$ is leaked to the adversary, but that will only benefit the adversary to generate the **next** batch of perturbed data.
> >
> > *attacking RMC under AA*. We mean attacking the initial static base model and transferring adversarial examples to RMC.
> >
> > *RMC vs. DENT*. DENT and RMC are two very different transductive defenses. RMC adapts the model based on a test data point x (possibly with adversarial perturbations) by fine-tuning the model on adversarial examples generated on training data that are close to x. DENT adapts the model on a test batch by minimizing the entropy with information maximization regularization. The fact that DENT is more robust than RMC under GMSA is because DENT uses the adversarially trained model as the base model and DENT only updates the batchnorm parameters. The adapted model from DENT is close to the adversarially trained model and thus it should be at least as robust as the adversarially trained model. Although RMC can also use an adversarially trained model as the base model, it updates the model aggressively and thus the adapted model may be far away from the original adversarially trained model. Thus, RMC's robustness is lower.
> >
> > **Q4**. For DENT, their defense relies on naturally ordered data, because DENT updates the model based on the test batch and by reordering data, we are able to construct different batches with label shifts, and it is our contribution that we found by reordering data, one can further break their defense. GMSA itself does not rely on ordering, but in the appendix we demonstrated that we can leverage order to get stronger attacks.

---

### Author Response · Authors · 2021-11-19
**Revised draft and we look forward to more questions if any**

We thank the reviewers again for the comments. Based on the comments (please refer to our individual replies for details), we have prepared a revised submission, where we focus on clarifying the most important confusions we identified. All revised texts are in blue.

1. We have revised the title of Section 5 as "Adaptive Attacks in One Round" to highlight that the attacks discussed in that section apply to attack one batch of test input (note in different rounds, independent batches are sampled). We also add a paragraph in the beginning to clarify, quote *"In this section we study a basic question: How to perform adaptive attacks against a transductive-learning based defense in one round? Note that, in each round of a multi-round game, an independent batch of test input $U$ is sampled, and the defender can use transductive learning to produce a model specifically adapted to the adversarial input $U'$ , after the defender receives it. Therefore, it is of fundamental interest to attack this ad-hoc adaptation. We consider white-box attacks: The attacker knows all the details of $\Gamma$, except private randomness, which is sampled after the attacker’s move."*

2. We have clarified how we extend to multi-round attacks: *"**Attacks in multi-round.** If the transductive-learning based defense is stateless, then we simply repeat one-round attack multiple times. If it is stateful, then we need to consider state-leaking setting or non-leaking setting. For all experiments in Section 6, we only evaluate non-leaking setting, which is more challenging for the adversary."*

3. We have revised the description of "Transductive Adversarial Training", and clarify that the experiment is about attacking an ensemble of models trained with independent randomness, quote: *"After receiving a set of examples at the test time, we always adversarially retrain the model using fresh randomness. The key point of this transduction is that private randomness is sampled after the attacker’s move, and so the attacker cannot directly attack the resulting model as in the inductive case. Specifically, for our GMSA attacks, we attack an ensemble of T = 10 models, adversarially trained with independent randomness, and generate a perturbed test set $U'$ . Then we adversarially train another model from scratch with independent randomness, and check whether $U'$ transfers to the new model (this thus captures the scenario described earlier). Somewhat surprisingly, we show that $U'$ does not transfer very well, and the TADV improves robustness significantly."*

4. We have emphasized why private randomness is modeled as not known to the adversary. See the paragraph "Private randomness" in Section 4, quote: *"Since these randomness are generated **after** the attacker’s move, they are treated as private randomness, and not known to the adversary."*

Please feel free to let us know if you have further concerns about our work. We believe that we have addressed all the issues raised.

---

### Decision · Program_Chairs · 2022-01-20

**Decision:**

Accept (Poster)

**Comment:**

The paper formalizes the adversarial attack problem for transductive defenses, where the model is sequentially updated with a batch of (adversarial) test inputs. The paper comes up with a quite generic attack scheme and their instantiation of this scheme shows that RMC and DENT are not robust respectively not more robust than the underlying adversarially robust base model.

Positive
- formal treatment of attacks on transductive defenses including discussion about different types of attacker knowledge
- the attack model is quite generic and could work for future transductive defenses and thus is a useful baseline attack which could be suggested to be used by future transductive defenses for robustness evaluation. In particular, as the standard AutoAttack is not designed for transductive defenses and thus can overestimate adversarial robustness

Negative
- the description is sometimes overly technical and some (important) details had to be clarified
- the technical novelty of the attack is limited
- the overall accuracy but also robust accuracy depends on the chosen batch. Therefore the authors should report mean and standard deviation over several different random draws of batches
- the Transductive Adversarial Training Defense seems to consist of adversarial retraining from scratch after each incoming batch. This is excessively costly and not practical.

Minor:
- The batch size is an important parameter which apparently is assumed to be known in this work

The paper is borderline. Two reviewers argue for rejection, two for acceptance. Only one reviewer engaged in the discussion.
In my point of view the positive point of having a reference for correct evaluation of adversarial robustness of transductive defenses weighs more than the raised negative points which can be fixed (at least partially). Thus I think that this paper is a valuable contribution to the field of adversarial robustness.